# The Threshold Effect of Urban Levels on Environmental Collaborative Governance: An Empirical Analysis from Chinese Cities

**DOI:** 10.3390/ijerph19073980

**Published:** 2022-03-27

**Authors:** Jie Fan, Zhuo Shen, Zhengwen Wang

**Affiliations:** 1School of Economics and Management, Wuhan University, Wuhan 430072, China; 2015101050003@whu.edu.cn; 2Information Institute of The Ministry of Emergency Management of PRC, Beijing 100029, China; shenzhuo@coalinfo.net.cn; 3National Institute of Insurance Development, Wuhan University, Ningbo 315100, China

**Keywords:** environmental collaborative governance, city hierarchy, threshold regression model, city scale

## Abstract

Taking 286 cities above the prefecture level in China as the research object and the research period from 2003 to 2019, through the two-way fixed effect model, this paper empirically analyzes the impact of the city level on environmental collaborative governance. The threshold regression model is used to reveal the nonlinear relationship between urban levels and environmental collaborative governance and to analyze the phase characteristics of different urban levels for environmental collaborative governance. The results show that: (1) The city level has a significant role in promoting collaborative environmental governance. (2) The regression results of the three major sections show that the urban level promotion in the eastern region has the strongest promoting effect on the coordinated environmental governance, and the western region is the weakest. (3) The impact of the city level on collaborative environmental governance is nonlinear. When the city scale reaches a certain critical point, its impact on the collaborative environmental governance tends to intensify.

## 1. Introduction

At present, China’s economy is in a special period of three superimposed periods of economic growth rate shifting, structural adjustment pains, and the digestion of previous stimulus policies. The quality of economic development is seriously threatened by environmental pollution [1,2]. In the final analysis, ecological and environmental issues are issues of development mode and way of life. To protect the ecological environment is to protect productive forces, and to improve the ecological environment is to develop productive forces [3,4]. The premise of environmental governance is to jointly protect the environment and promote the sustainable development of China’s economy [5,6,7]. For a long time, various regions of China have been governed separately without forming a unified whole. Due to the vicious competition for resources and the homogeneous and disorderly competition of industries, the ecological resources have been severely damaged. First, in the existing environmental management system and mechanism, cross-regional cooperation governance is difficult, the concept of regional cooperation has not yet been formed, and environmental governance is independent. Moreover, territorial management is overemphasized, environmental law enforcement is divided into regions, and cross-regional illegal solid hazardous waste dumping is common [8,9]. Second, industrial transfer is determined by the stage of economic development. Industrial transfer often hides pollution transfer. With the continuous economic development, high-energy-consuming industries in the eastern region are gradually transferred to the central and western regions, which will bring pollution transfer and diffusion. Third, pollution has a certain inertia, and it is difficult to completely counteract the large amount of pollutants discharged in a short period of time, and the potential hidden dangers of environmental risks are still increasing [10,11,12]. Finally, unregulated competition in various regions leads to low resource utilization efficiency, a lack of collaborative spirit, and the phenomenon of land and resource grabbing which aggravates regional divisions making it difficult to form plans for environmental governance [13,14,15].

The key to environmental governance lies in collaborative governance. Synergy is the study of cooperation. The theory of synergy regards the research object as a system composed of several subsystems, and the subsystems interact in various ways to make the whole system produce a synergy [16,17,18]. The theory of synergy holds that in a stable system, - all subsystems are coordinated in a certain way and move together. The contribution of synergistic organization is to establish a research framework for dealing with complex systems with a unified view. The synergy theory is suitable for solving the problems existing in China’s environmental governance. If the environmental governance of Chinese cities is regarded as a whole system, then the environmental governance capacity of each city is a subsystem, and the problem that needs to be solved is the collaborative environmental governance among cities [19,20,21]. To make the overall system of environmental governance in China operate stably, that is, to effectively manage environmental problems, cities need to govern environmental problems collaboratively. It means that cities should cooperate with each other, participate in environmental work according to certain regulations, form synergies, and consider the long-term interests of each city, ultimately achieving effective ecological environment governance, environmental protection, and long-term stability, which is what the process of environmental governance is. The research of domestic and foreign scholars on environmental collaborative governance focuses mainly on evaluating environmental collaborative governance and conducting in-depth discussions on how to optimize the allocation of innovative resources and improve environmental collaborative governance in various regions [22,23]. These studies focus mainly on measuring and evaluating regional collaborative environmental governance, depicting the temporal and spatial evolution process of collaborative environmental governance and revealing its impact mechanism. In contrast the research on the relationship between urban levels and collaborative environmental governance is less involved.

Urban hierarchy refers to a ranking structure formed by taking cities within a country based on city size or administrative hierarchy [24,25]. Innovation is also one of the important factors in the composition of the urban system [26,27,28,29]. In fact, there is a clear overlap in the spatial distribution of urban levels and urban innovation levels. For China, in the context of fierce competition among local governments, the difference in urban administrative levels will affect the aggregation of urban innovation resources to a certain extent, and high-level cities often have high administrative levels. As a result, it will also affect the agglomeration of innovation, the construction of regional environmental scientific research and technology platforms, and the improvement of environmental protection technology. High-level cities are more conducive to promoting green and low-carbon development such as optimizing the green development pattern, upgrading the industrial structure, optimizing the energy structure, coping with climate change, and practicing green and low-carbon lifestyles. In addition, environmental infrastructure such as high-level urban sewage collection and treatment facilities, port environmental facilities, regional environmental emergency response capabilities, and ecological environment monitoring systems are well-established, which is conducive to serving surrounding cities and forming environmental collaborative governance.

Obviously, environmental governance is closely related to urban needs. Cities are the spatial carriers of high-quality production factors, and large cities play a leading role in green development and green lifestyles. However, as has been mentioned above, the existing research about environmental collaborative governance always concentrates on the measurement of collaborative environmental governance and its impact mechanism. It rarely deals with the gap in the relationship between urban level and environmental collaborative governance. Does the city level constitute a binding condition for collaborative environmental governance? If so, what mechanism is used to influence collaborative environmental governance? Under different city scales, is there a nonlinear relationship between city level and environmental cogovernance? Is there an optimal city level to promote the improvement of environmental collaborative governance? In response to the above problems, this research will take 286 cities above the prefecture level in China as the research object, conduct a systematic study on the relationship between the city level and the environmental collaborative governance, and empirically analyze the impact of the city level on the environmental collaborative governance through a two-way fixed effect model. On this basis, the panel threshold model is used to estimate the nonlinear relationship between the city level and the urban environmental collaborative governance. We explore the path differences and phase characteristics of the environmental collaborative governance of cities of different scales affected by the city level to provide information for the construction of an environment-friendly society. 

## 2. Econometric Model Construction and Variable Selection

### 2.1. Model Construction

#### 2.1.1. Panel Model

Based on the above analysis, this paper focuses on revealing the impact and degree the city level has on the urban environment collaborative governance in the empirical part and establishes the following panel least squares regression model:(1)$${Envir}_{it}\;=\;\beta_{1}{Urbh}_{it}\;+\;\beta X_{it}\;+\;\beta_{0}\;+\;T\;+\;\alpha \;+\;\varepsilon_{it}$$

In the formula, $Envir_{it}$ represents the urban environment collaborative governance,$\;Urbh_{it}$ represents the city level, $X_{it}$ is the control variable, *T* is the time fixed effect, ***α*** is the individual fixed effect, $\varepsilon_{it}$ is the random interference item, and *β*, *β*_0_, *β*_1_ is the coefficient to be estimated.

#### 2.1.2. Panel Threshold Model

Threshold regression tests whether the parameters of the sample groups divided according to the threshold value are significantly different, and it is used to study the heterogeneity of the interaction between variables. The threshold regression model developed by Hansen (1999) [30] can endogenously divide the data interval according to the characteristics of the data itself, avoiding the randomness of artificially dividing the sample interval. There may be a nonlinear relationship between each city level and environmental collaborative governance. Traditional linear regression cannot explain the relationship between the two well. The threshold model regression is more realistic. Therefore, this paper adopts the threshold regression model of Hansen (1999) [30], taking the total urban GDP as the threshold variable, and first setting the following single threshold regression model:(2)$${\mathrm{Envir}}_{\mathrm{it}}\;=\;\lambda_{0\;}+\;\lambda_{1}D_{it}\cdot {I\;(\mathrm{Gdps}}_{\mathrm{it}}\;\leq {\;r}_{1})\;+\;\lambda_{2}D_{\mathrm{it}}\cdot I\;(Gdp_{\mathrm{it}}{\;>\;r}_{1})\;+\;\lambda_{3}X_{\mathrm{it}}\;+\;\gamma \cdot {t\;+\varepsilon }_{\mathrm{it}}$$
where I(⋅) represents the indicative function, which takes the value 1 when the expression in the parentheses is true and 0 when it is false. *D_it_* is the core explanatory variable, $Gdp_{it}$ is the threshold variable, *X_it_* is the control variable, *ε_it_* is the random disturbance term. When $Gdp_{it}\;$≤ *r*_1_, the core explanatory variable *D_it_* coefficient is *λ_1_*, when $Gdp_{it}\;$> *r*_1_, the core explanatory variable *D_it_* coefficient is *λ_2_*, t is the time effect, *λ* is a constant term, *ε_it_* ~(0, *σ*) is a random interference term. The similarities and differences between *λ_1_* and *λ_2_* are what we focus on.

Equation (2) only assumes that there is one threshold, but there may be two or more thresholds. Due to space limitations, the test of two or more thresholds will not be repeated in this paper.

### 2.2. Variable Selection and Data Sources

#### 2.2.1. Urban Environment Collaborative Governance Capability (Envir)

Ecological environmental protection and economic development are not a relationship of contradiction and opposition, but a relationship of dialectical unity. The success or failure of environmental protection depends on the final analysis of the economic structure and the mode of economic development. Therefore, this paper refers to relevant research [31,32] to construct an evaluation index system for Chinese urban green development and environmental governance that integrates economic development and environmental governance. Ecological economy refers to the development of economically developed and ecologically efficient industries within the carrying capacity of the ecosystem. The industrial structure can better reflect the development level and development stage of the region. Especially with the increase in the proportion of tertiary industry, the regional industrial structure tends to be reasonable and optimized, which will effectively improve the local environmental governance capacity. Therefore, in this paper, the proportion (%) of the tertiary industry in GDP is used to represent the development level of the ecological economy. Energy consumption refers to the energy consumed for production and living. Energy consumption per capita is an important indicator to measure a country’s economic development and people’s living standards. The more energy consumption per capita, the greater the gross national product and the richer the society. In developed countries, changes in energy consumption intensity are closely related to the industrialization process and environmental governance. The power consumption per CNY 10,000 of GDP (kWh/person) is used to reflect the effect of energy-saving policies and measures and to test the effect of energy-saving and consumption reduction. Pollution discharge is measured by sewage discharge per unit of industrial output value (ton/CNY 10,000); investment in environmental governance is the mainstay, reflecting the source participation and whole-process control of environmental governance. Investment in environmental protection fixed assets is an effective means to improve environmental quality. The proportion of investment in environmental protection in GDP is an important indicator to measure the ability of a certain region to protect the environment. The proportion of investment in fixed assets in environmental protection in GDP (%) reflects the environmental governance of each city of importance. Governance effectiveness (E) focuses on the treatment rate of pollutants and the status quo of the regional ecological environment. The urban sewage treatment rate (%) reflects the regional sewage treatment effect and represents the environmental treatment capacity of each city [33,34].

The method of combining the subjective and objective weights of the comprehensive principal component analysis method and the entropy value method gives the weights. The weights assigned to the above five indicators in this paper are 0.1178, 0.2341, 0.1524, 0.1832, and 0.3125. The coupling degree C_AB_ of the environmental collaborative governance capacity between the two cities of AB is expressed by formula refers to Sun et al. (2021) [8].

#### 2.2.2. Urban Level (*Urbh*)

This paper refers to Wang et al. (2019) [35] to construct a city-level index evaluation system from the aspects of consumption scale, resource energy consumption level, degree of opening to the outside world, manufacturing level, employment scale, and infrastructure level. Due to the significant linear relationship between the city’s administrative level and the existing indicators, it is not included in the city-level indicator system. Finally, this paper selects the total retail sales of urban consumer goods (CNY 10,000), the total electricity consumption of the whole society (10,000 kWh), the per capita electricity consumption (kWh/person), actual use of foreign capital (USD 10,000), the proportion of actual use of foreign capital in the regional GDP (%), the number of employees in the tertiary industry (10,000 people), the passenger volume of civil aviation (10,000 person-times) [36,37], the proportion of civil aviation passenger traffic to the population (%), and the proportion of road and water passenger traffic to the population (%) to evaluate the city level. Since this paper takes the city level as the core explanatory variable, in addition to testing the correlation, it will also examine whether the city level has a causal relationship with the collaborative environmental governance [38,39,40,41]. Therefore, the city level cannot only be assigned and ranked, but also the impact of changes in the city level on the collaborative environmental governance can be measured. Using the subjective and objective weight method, the final weights given to the above nine indicators in this paper are 0.0753, 0.0962, 0.1127, 0.1356, 0.2125, 0.1402, 0.1522, 0.0325, and 0.0428.

#### 2.2.3. Control Variable

In addition to the explanatory variables above, there may be variables that do not appear in the model but that influence the explained variables. To improve the accuracy of the model estimation in this paper, control variables must be added. Control variables have two main functions: one is that they may contain other factors that affect the synergy of environmental governance, which helps to alleviate endogeneity problems; the other is that they can characterize regional characteristics and isolate regional heterogeneity as much as possible [42,43,44,45]. The control variables in this paper mainly include energy consumption, green area, road area, and the number of industrial enterprises. These variables are directly or indirectly related to environmental pollution [46,47,48,49,50].

This paper takes 286 cities at the prefecture level and above in China as the research object, and the data comes from the *China Urban Statistical Yearbook* (2004–2020) because of data availability.

## 3. Empirical Results and Analysis

### 3.1. Benchmark Regression Results

After the F-test of the “city” dummy variable, the individual effect exists at the 1% significance level, and the null hypothesis that there is no individual effect is rejected. The model considering the individual effect is divided into fixed effect and random effect models. After the Hausman test, the *p*-value is close to 0, so the null hypothesis of choosing a fixed-effect model is acceptable. In summary, the regression equation constructed in this paper is a two-way fixed-effect model. First, the least squares regression is adopted. Columns (1) and (2) of Table 1 are the regression results of the two-way random effect and fixed effect of the benchmark model, respectively. Due to the large variance of the urban environmental collaborative governance index, it is necessary to delete more to ensure the robustness of the regression results. Column (3) of Table 1 is the regression result after removing the top 10% of samples of urban environmental collaborative governance.

In Table 1, in Models (1)–(3), the city level has shown a significant role in promoting the level of collaborative environmental governance. It can be found that the city level has a significant positive impact on the collaborative governance of the urban environment: in Models (1) and (2), and every percentage point increase in the city level variable can increase the collaborative governance of the urban environment by about 0.25 units, The sign and significance of the key explanatory variables in the regression results without extreme values have not changed, indicating that under different conditions, the role of the city level on environmental collaborative governance is not covered by other influencing factors. It illustrates the importance of the city level to collaborative environmental governance. The possible reason is that the economic development model of low-level cities is based mainly on the extensive economy, the environmental pollution is serious, and the level of environmental collaborative governance is relatively low; the environmental collaborative governance index also began to pick up. Due to their superior infrastructure construction and innovative cultural atmosphere, high-level cities often have more advanced technology and environmental protection awareness, which is conducive to the improvement of environmental collaborative governance capabilities. High-level cities have passed the pollution stage in the early years, and now high-tech nonpolluting industries have developed. In contrast, low-level cities are still in the stage of high-polluting industries cannot be developed overnight. Industrial development has a gradual process from low to high, and it is impossible to skip a certain stage. Environmental protection policies should consider the different scales and characteristics of different cities, where their industries are in the cycle of pollution and choose governance measures suitable for the city level according to local conditions.

### 3.2. Robustness Check

For the robustness test, this paper will test the robustness of the regression results from the perspectives of geographic heterogeneity and lag effect.

First, we consider the impact of geographic heterogeneity on the results. This paper uses block regression to test the robustness. Columns (1)–(3) in Table 2 are the regression results for eastern, central, and western China, respectively. Through comparative analysis, in the regression results of the three major sectors, the influence coefficients of the city level on the collaborative environmental governance are all positive, and the significance level is 1%, which is enough to show that the influence of geographical heterogeneity on the conclusions of the article is not enough. Among them, the promotion of the urban level in the eastern region has the greatest effect on environmental collaborative governance compared with other regions (the coefficient is 42.185), followed by the central region (the coefficient is 30.258), and the western region is the smallest (the coefficient is 15.856). There is heterogeneity in the promotion of collaborative governance among the three sectors. The possible reasons for this phenomenon are that the eastern region has a relatively high level of economic development, advanced technology, better industrial transformation and upgrading, and a strong awareness of environmental protection. Therefore, the improvement of the city level will help cities play a leading and demonstrative role in environmental collaborative governance. However, the promotion of urban tiers in the western region has the lowest effect on the coordinated environmental governance among the four major sectors. This may be due to economic development and industrial transformation., while there is much improvement needed to upgrade the western region. Therefore, it is difficult for the hierarchy to play a significant role in promoting the coordinated environmental governance. It also shows that the development of the western region is urgent, and it is particularly necessary to focus on optimizing the urban environmental governance capacity.

Second, we consider the hysteresis effect. The collaborative environmental governance between cities is not an instantaneous behavior. It may take some time for the improvement of the city level to have an impact on the collaborative environmental governance. The urban level reflects the long-term variation of urban development, while the collaborative environmental governance capability reflects short-term fluctuations. The direct regression of the two may have estimation bias. Therefore, to avoid this situation, this paper re-regresses all explanatory variables with a lag of one period. As shown in column (4) of Table 2, there is no fundamental change in the positive promotion of urban environmental cogovernance at the city level.

## 4. Threshold Effect Test

City scale is often recognized as an important factor affecting the effectiveness of urban environmental pollution control, and it is closely related to the sustainable growth of the urban economy. The same measures of environmental governance usually show different effects in cities of different scales. To verify the heterogeneity of the effect of city level on the collaborative governance of an urban environment under different scales, this paper selects city scale as the threshold variable to perform panel threshold model regression. This paper uses GDP as a proxy variable for city size. The results of the threshold effect test are shown in Table 3. It can be found that the single threshold model has the highest F value when GDP is used as the threshold variable. Therefore, this paper selects the single threshold model. When the single threshold variable GDP is used as the threshold variable, the threshold value is 1.566×10^7^.

As shown in Column (3), GDP as a threshold variable fully exerts the effect of a single threshold. After the urban GDP exceeds CNY 156.6 billion, the promotion of urban tiers has significantly improved the promotion of environmental collaborative governance. In practice, exceeding this threshold, the sample accounted for roughly 23.2% of the total sample. With the improvement of the level of urban economic development, the role of the city level in agglomeration of innovative elements and environmental resources has become more and more obvious (Table 4). Therefore, the role of the city level in promoting the collaborative governance of the urban environment has gradually increased. The reason is that when the level of economic development is low, urban development is often in a state of blind and disorderly expansion. Although the urban level is improved, the corresponding urban functions and supporting measures are not suitable for the improvement of the urban level. However, when the level of economic development reaches a certain level, the process of industrialization and modernization of the city is accelerated. With the increase of the per capita income of urban residents, the requirements for environmental quality are rising higher and higher, and the urban economic development needs to realize the transformation from factor-driven to innovation-driven as soon as possible. At the same time, the upgraded city level will be able to attract a large number of talents and improve urban pollution treatment technology and environmental governance capabilities more effectively, thereby improving urban environmental collaborative governance more effectively. As a result, the influence coefficient of *Urbh* is only 5.632 when the threshold variable GDP is less than 1.566×10^7^, and then the influence coefficient of *Urbh* rises to 48.235 after urban GDP reaches 1.566×10^7^. This also shows that cities at the top of the pyramid have a more obvious demand for environmental governance. Only when cities reach a certain scale level can they have extensive radial power. Core cities need to be supported by corresponding economic strength to better drive the surrounding urban environment, leading to overall improvement.

## 5. Conclusions and Policy Recommendations

(1) The larger the urban scale and the higher the rank of Chinese cities, the higher the coordinated environmental governance. The city level and the environmental collaborative governance are measured respectively according to the city level index system and the environmental collaborative governance index system. The measurement results show that the improvement of the city level can significantly improve the environmental collaborative governance. Since the city level is closely related to environmental collaborative governance, the higher the city level, the denser the human capital, the more perfect the service industry system, and the stronger the environmental governance capability. The improvement of the city level is conducive to the environmental collaborative governance, and the environmental collaborative governance has a scale effect. Therefore, the environmental collaborative governance capacity matches the city level. For the country, the role of high-level cities in regional environmental collaborative governance should be brought into play, closely linked to the regional integrated development and the joint protection of the ecological environment, and formulated to form the division of labor, complementary advantages, and overall planning. The action cogovernance and joint protection plan gives full play to the comparative advantages of high-level cities in green development and green lifestyles and establishes a central area for coordinated treatment of environmental pollution around high-level cities. In addition, the plan gives full play to the radiation role of high-level cities and promotes coordinated treatment of regional environmental pollution, continuously improving the quality of the ecological environment. In this way a regional ecological environment and coordinated protection supervision system can improve the collaborative governance mechanism of regional environmental pollution. The regional ecological environment can integrate the protection and governance mechanism collaboratively, finally realizing the integrated governance of the regional environment.

(2) The results of subregions show that the urban level in the eastern region has the most significant role in promoting environmental collaborative governance, followed by the central region and the western region. This shows that in the coordinated environmental governance of Chinese cities, each region should adapt to local conditions, give full play to the comparative advantages of each region, strengthen the design of regional and cross-border quantitative indicators, and promote the continuous reduction of energy consumption per unit of GDP and carbon dioxide emissions. We need to pay attention to the coordinated advancement of environmental governance and ecological protection, strictly control the red line of ecological protection, basically form a cross-regional and cross-basin ecological network, effectively protect biodiversity, steadily enhance the ecosystem service function, and continuously improve the supply capacity of high-quality ecological products.

(3) Threshold regression results show that only when a city reaches a certain scale level can it have a wider radiating power and drive the environmental collaborative governance capacity of surrounding cities. The threshold regression results show that the improvement of the level of the second top city has the most obvious role in promoting environmental collaborative governance. Therefore, the key to coordinated regional environmental governance is to coordinate the promotion of regional green development layout, structural adjustment, and lifestyle changes. To accelerate the transformation, upgrading and layout adjustments of high-pollution, high-emission, and high-risk industries must be undertaken to optimize the energy structure and promote some regions and industries to take the lead in achieving peak carbon emissions. It is necessary to focus on key links and solve outstanding problems. From the perspective of regional integration, these key systematic regional and transboundary ecological and environmental problems should be addressed in a coordinated manner.

The above results offer a scientific reference for governments formulating environmental regulation policy, so this research is of great significance to improve environmental government efficiency, raise life quality of the people, and contribute to achieve society’s sustainable development. At the same time, future research can be developed with microlevel data or by using more kinds of econometric methods. In addition, the comparison among different countries is also worth studying.

## Figures and Tables

**Table 1 ijerph-19-03980-t001:** Benchmark regression results.

	(1) *Re*	(2) *Fe*	(3) Exclude Extreme Values
*Urbh*	25.321 *** (15.28)	25.452 *** (12.98)	15.782 *** (15.24)
*Gas*	−0.328 *** (−5.08)	−0.318 *** (−5.01)	−0.252 *** (−4.28)
*Green*	0.128 * (2.18)	0.152 (1.25)	0.056 ** (2.18)
*Road*	1.328 (0.86)	0.685 (0.41)	0.214 (1.32)
*Ent*	−7.02 (−0.98)	−5.52 (−1.23)	−4.29 *** (−2.09)
*Constant*	5.235 *** (2.56)	3.215 (0.63)	0.758 ** (1.75)
*R^2^*	0.4251	0.4562	0.4788

Note: t values are in parentheses; ***, ** and * indicate significance at the 1%, 5% and 10% levels, respectively. The result is calculated by Stata 14.0, the same below.

**Table 2 ijerph-19-03980-t002:** Robustness test results.

	(1) Eastern China	(2) Central China	(3) Western China	(4) One Period Behind
*Urbh*	42.185 *** (12.35)	30.258 *** (15.28)	15.856 *** (7.48)	26.152 *** (13.23)
*Gas*	−0.286 *** (7.38)	−0.029 (0.189)	−0.254 *** (7.52)	−0.251 *** (7.20)
*Green*	0.125 * (1.38)	0.069 (0.125)	0.055 *** (0.152)	0.032 *** (0.125)
*Road*	11.572 *** (3.73)	4.211 *** (1.42)	5.242 (1.30)	6.352 *** (2.25)
*Ent*	−0.852 (−1.22)	0.125 (0.134)	−0.212 (−0.125)	0.252 (0.231)
*Time Effect*	*control*	*control*	*control*	*control*
*Individual Effect*	*control*	*control*	*control*	*control*
*Constant*	5.231 ** (2.25)	4.562 *** (2.56)	3.956 ** (2.11)	4.956 *** (3.22)
*R* ^2^	0.6232	0.5685	0.4025	0.4521

Note: t values are in parentheses; ***, ** and * indicate significance at the 1%, 5% and 10% levels, respectively.

**Table 3 ijerph-19-03980-t003:** Threshold effect test.

*Threshold Variable*	*Threshold Number*	*F*	*P*	1% *Threshold*	5% *Threshold*	10% *Threshold*
GDP	*single threshold*	365.232 ***	0.000	38.523	21.356	14.212
	*double threshold*	92.358 ***	0.000	20.325	12.356	7.523
	*three thresholds*	0.000 *	0.085	0.000	0.000	0.000

Note: t values are in parentheses; *** and * indicate significance at the 1%, 5% and 10% levels, respectively.

**Table 4 ijerph-19-03980-t004:** Threshold effect analysis.

*Urbh*	threshold variable <*δ*_1_	5.632 *** (3.25)	*Gas*	−0.221 *** (−4.52)
	*δ*_1_ ≤threshold variable <*δ*_2_	48.235 *** (23.24)	*Green*	0.018 *** (2.52)
			*Road*	3.256 *** (2.21)
	*δ*_2_ ≤threshold variable	-	*Ent*	−2.25 (−1.74)
*Time Effect*		*Control*	*Individual Effect*	*Control*
*Constant*		2.752 ** (2.34)	R^2^	0.5428
*δ* _1_		1.566 × 10^7^	*δ* _2_	-

Note: t values are in parentheses; *** and ** indicate significance at the 1%, 5% and 10% levels, respectively.

## Data Availability

No new data were created or analyzed in this study. Data sharing is not applicable to this article.

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
