# Peer review of "The Threshold Effect of Urban Levels on Environmental Collaborative Governance: An Empirical Analysis from Chinese Cities"

_ijerph, 2022, doi:10.3390/ijerph19073980_

Round 1

Reviewer 1 Report

Line 55 - 78 – The authors underline the necessity of collaborative governance. Nevertheless, the term "environmental governance" is abused but the notion of this process and components related to environmental deficits in selected cities is not defined. Moreover the sentence „These studies mainly focus on the measurement and evaluation of regional collaborative environmental governance, depicting the temporal and spatial evolution process of collaborative environmental governance, and revealing its impact mechanism, but the research on the relationship between urban levels and collaborative environmental governance is less involved” need to be clarified.

Line 105 – what were the criteria to choose mentioned: „286 cities above the prefecture level in China" as the research object?

Line 108 – the sentence „two-way fixed effect model. effect.” need to be corrected. Similarly:

Line 155 – „..governance. aspects to quantify” 

Line 218  - „…on the collaborative environmental governance. Influence” 

Line 371 – „…carbon dioxide emissions. work guidelines..” 

Line 360-364 the sentence formulated as a result of findings „Give full play to the leading role of high-level cities in green development and green lifestyles, continuously improve the quality of the ecological environment, basically establish a regional ecological environment coordinated protection supervision system, and improve the regional ecological environment integrated protection and governance mechanism” they seem to be based on the beliefs of the authors and not the results of the analyzes carried out.

The analytical model (Panel Threshold Model) is based on a very general set of variables that applies to 286 selected cities. The aim of the study is to analyze the heterogeneity of interactions between variables. The authors of this article chose the city scale as the threshold variable to perform a panel regression of the threshold model. There is a need to better justify why this variable has been selected by disregarding other elements that are inherent in environmental management. Without even basic information on what environmental challenges exist in these selected cities, some conclusions seem to be very general and strongly affected by the above assumption. It is recommended to include at least some kind of concentration of joint environmental management activities in these cities. This may then open up the conditions for a critical revision of the results, which is currently quite limited.

Author Response

  1. Line 55 - 78 – The authors underline the necessity of collaborative governance. Nevertheless, the term "environmental governance" is abused but the notion of this process and components related to environmental deficits in selected cities is not defined. Moreover the sentence „These studies mainly focus on the measurement and evaluation of regional collaborative environmental governance, depicting the temporal and spatial evolution process of collaborative environmental governance, and revealing its impact mechanism, but the research on the relationship between urban levels and collaborative environmental governance is less involved” need to be clarified.

A: As you mentioned that the notion of environmental governance and components related to environmental deficits in selected cities is not defined, the sentences in Line 55- 78 have been revised to make the notion of “environmental governance” more apparent, but as for the components related to environmental deficits related environmental deficits in selected cities, what does it mean? Does it mean the correct measures used in the process of environmental governance that can improve the deficits in selected cities? If so, it has been defined in the notion of environmental governance now. Then, you proposed that the sentence “These studies mainly focus on ……. Is less involved” need to be clarified. However, this sentence is just a summary of previous work for the similar topic, and it has been summarized concise. It seems that it doesn’t need to be further clarified.

  1. Line 105 – what were the criteria to choose mentioned: „286 cities above the prefecture level in China" as the research object?

A: The 286 cities above the prefecture level in China is commonly choose because prefecture level is more micro than provincial level, and the data of these cities is relatively complete and obtainable. The criteria have been added in the end of control variable section.

  1. Line 108 – the sentence „two-way fixed effect model. effect.” need to be corrected. Similarly:

Line 155 – „..governance. aspects to quantify” 

Line 218  - „…on the collaborative environmental governance. Influence” 

Line 371 – „…carbon dioxide emissions. work guidelines..” 

A: Thank you for your carefully check, these are mistake writings, now they have been corrected.

  1. Line 360-364 the sentence formulated as a result of findings „Give full play to the leading role of high-level cities in green development and green lifestyles, continuously improve the quality of the ecological environment, basically establish a regional ecological environment coordinated protection supervision system, and improve the regional ecological environment integrated protection and governance mechanism” they seem to be based on the beliefs of the authors and not the results of the analyzes carried out.

A: As the last section of this paper is entitled as “Conclusions and policy recommendations”, the sentences you mentioned is about the policy recommendations, as you said, it seems not to be very closed to the results of the analyzes carried out, but the main result is that the cities with higher urban level have a comparative advantage in environmental governance capability, so the sentence is not only based on the beliefs but also related to the results. To make the sentence more closed to the results, it has been made some adjustments.

  1. The analytical model (Panel Threshold Model) is based on a very general set of variables that applies to 286 selected cities. The aim of the study is to analyze the heterogeneity of interactions between variables. The authors of this article chose the city scale as the threshold variable to perform a panel regression of the threshold model. There is a need to better justify why this variable has been selected by disregarding other elements that are inherent in environmental management. Without even basic information on what environmental challenges exist in these selected cities, some conclusions seem to be very general and strongly affected by the above assumption. It is recommended to include at least some kind of concentration of joint environmental management activities in these cities. This may then open up the conditions for a critical revision of the results, which is currently quite limited.

A: Thank you for your valuable suggestion, you mentioned that there is no information about the 286 selected cities about the environmental challenges exist in these areas, but the first paragraph is about the environmental challenges in these cities, such as “Due to the vicious competition …… Dumping is frequent.” You may view that the 286 selected cities are some special samples in China, actually they are all routine samples to reflect China’s problem. So, the environmental challenges about China mentioned in the introduction section are just what the 286 cities are faced to. Besides, selecting the city scale as threshold doesn’t mean disregarding other elements that are inherent in environmental management, it is mainly because the interest of this research is to verify the heterogeneity of the effect of city level on the collaborative governance of urban environment under different scales. To make the reason for selecting city scale as the threshold variable clearer, some sentences has been added in the beginning of the threshold effect test section. And as you suggested, some kind of concentration of joint environmental management activities has been included in the suggestion part now.

  Thank you for your warm work and valuable suggestions earnestly. Hoping the correction will meet with approval.

Reviewer 2 Report

The article presented for review is correct in terms of its content. This is an interesting article. It deals with topical issues. The article correctly analyzes the influence of the city level on environmental management. The adopted model is adequate. Conclusions are well exposed against the background of the analysis. Proper selection of literature. These are the positive aspects of the study. Potentials for a possible (non-obligatory) improvement of the article:
-) Do the authors have information about other cities outside of China? Limiting the research to one country (large in terms of area) makes sense, but the research comparing different cultures, laws, and environmental traditions looks much better. If it is possible, I suggest inserting a few sentences from the literature review to the introduction.
-) The research results come from the period 2003-2016. These are slightly distant times from today. Everything is changing faster and faster when it comes to environmental management. I propose to add an explanation of the research period used.
-) I suggest not to end the subsection with a table such as tab. 1 in subsection 3.2 or tab. 2 in subsection 3.3. I propose to end the subsection with a summary text.
-) In conclusion, I would suggest to give a few sentences of the practical application of the obtained research results.
-) I also propose to propose further research directions.
I don't feel qualified to judge about the English language and style.
I recommend that you publish the article after making these minor improvements.

Author Response

The article presented for review is correct in terms of its content. This is an interesting article. It deals with topical issues. The article correctly analyzes the influence of the city level on environmental management. The adopted model is adequate. Conclusions are well exposed against the background of the analysis. Proper selection of literature. These are the positive aspects of the study. Potentials for a possible (non-obligatory) improvement of the article:

-) Do the authors have information about other cities outside of China? Limiting the research to one country (large in terms of area) makes sense, but the research comparing different cultures, laws, and environmental traditions looks much better. If it is possible, I suggest inserting a few sentences from the literature review to the introduction.

-) The research results come from the period 2003-2016. These are slightly distant times from today. Everything is changing faster and faster when it comes to environmental management. I propose to add an explanation of the research period used.

-) I suggest not to end the subsection with a table such as tab. 1 in subsection 3.2 or tab. 2 in subsection 3.3. I propose to end the subsection with a summary text.

-) In conclusion, I would suggest to give a few sentences of the practical application of the obtained research results.

-) I also propose to propose further research directions.

I don't feel qualified to judge about the English language and style.

I recommend that you publish the article after making these minor improvements.

A: Thank you very much for your recognition and valuable suggestion for the paper. Each of your comments has been considered carefully and the replay is as follows:

  1. Do the authors have information about other cities outside of China? Limiting the research to one country (large in terms of area) makes sense, but the research comparing different cultures, laws, and environmental traditions looks much better. If it is possible, I suggest inserting a few sentences from the literature review to the introduction.

A: As you suggested, the comparation among different cultures, laws, and environmental traditions will actually make the research more sense in a comprehensively form, but as the research country is China, the relevant literature is also mainly about China, so it’s very difficult to add the information about other cities outside of China. Hoping you can understand.

  1. The research results come from the period 2003-2016. These are slightly distant times from today. Everything is changing faster and faster when it comes to environmental management. I propose to add an explanation of the research period used.

A: Actually, the research results come from the period 2003-2019. The period 2003-2016 in the abstract is a mistake. As the data source in 2.3.3 section mentioned, the data is from the "China Urban Statistical Yearbook" (2004-2020), so the data is actually close to the current period.

  1. I suggest not to end the subsection with a table such as tab. 1 in subsection 3.2 or tab. 2 in subsection 3.3. I propose to end the subsection with a summary text.

A: In fact, the subsections 3.2 and 3.3 doesn’t end with a table such as tab.1 and tab.2, the table is shown in the end of the section just because the table is always shown after analysis and summary text. In other words, the summary text is included in the text above those tables.

  1. In conclusion, I would suggest to give a few sentences of the practical application of the obtained research results.

A: As you suggested, a few sentences about practical application have been added in the last paragraph of the article.

  1. I also propose to propose further research directions.

A: As you suggested, some further research directions have been also added in the last paragraph of the article after brief introducing the practical application of the research results.

  Thank you for your warm work earnestly. Hoping the correction will meet with approval.

Reviewer 3 Report

This manuscript examines factors that influence environmental collaborative governance.  The data set comes from China.  The authors are interested in determining whether there are threshold levels of items such as city scale (with GDP as the operative proxy) which influence the effect of the independent variables in a multiple regression on the dependent variable.  This is an interesting research question, and one that has potentially important policy implications. 

The manuscript is well organized and well laid out, The use of subheadings, paragraph breaks and tables are all appropriate.  The authors also are very frank about the political and competitive realities in China that may impinge on the topic at hand, which they describe in the introduction.

The authors use sound regression methods and seem to interpret their findings correctly.  The items in section 5 (Conclusions) follow from the results.

Decision: Accept

Reviewer 4 Report

The article entitled "The Threshold Effect of Urban Levels on Environmental Collaborative Governance: An Empirical Analysis from Chinese Cities" is well written in clear and understandable way. The methodology is clearly explained.

My main question to the authors is about the aim and hypothesis itself. Is it not the research question quite obvious? I mean, did you expect another result?

In my opinion that is obvious, that only more developed cities think about the ecological aspects of development and polices. The ecology is higher need, so only the richer and more self-conscious units and societies think of the ecology.

In my opinion the research is not enough based in the previous literature of the subject. Please, prove that your research answer for a research gap base on the state of the art. 

Technical checking. Please read the article in order to technical aspect and language. For example: line 108 there is a useless "effect", line 382, there is a dot after space. 

Author Response

The article entitled "The Threshold Effect of Urban Levels on Environmental Collaborative Governance: An Empirical Analysis from Chinese Cities" is well written in clear and understandable way. The methodology is clearly explained.

My main question to the authors is about the aim and hypothesis itself. Is it not the research question quite obvious? I mean, did you expect another result?

In my opinion that is obvious, that only more developed cities think about the ecological aspects of development and polices. The ecology is higher need, so only the richer and more self-conscious units and societies think of the ecology.

In my opinion the research is not enough based in the previous literature of the subject. Please, prove that your research answer for a research gap base on the state of the art. 

Technical checking. Please read the article in order to technical aspect and language. For example: line 108 there is a useless "effect", line 382, there is a dot after space.

A:  Thanks for your recognition for the paper. As you suggested, the relevant literature in the second paragraph has been make some updates (Line 74, Literature 22-23). Besides, to protrude the research gap based on the state of art more clearly, some words in the last paragraph of the introduction section has been added (Line 98 – Line 101). Then, thank you very much for your careful checking, now the similar errors have been checked again and corrected.

Once again, thank you very much for your valuable suggestions

Round 2

Reviewer 4 Report

Tank to authors for their improvements of the paper. The article is quite good. I think is suitable to publish, however there is always something to be done better. The problem is interesting, but I think could be better based on the previous research and literature.